# The Mediating Effect of Cytokines on the Association between Fungal Sensitization and Poor Clinical Outcome in Asthma

**DOI:** 10.3390/biomedicines10061452

**Published:** 2022-06-19

**Authors:** Ching-Hsiung Lin, Yi-Rong Li, Chew-Teng Kor, Sheng-Hao Lin, Bin-Chuan Ji, Ming-Tai Lin, Woei-Horng Chai

**Affiliations:** 1Division of Chest Medicine, Department of Internal Medicine, Changhua Christian Hospital, Changhua 500, Taiwan; 112364@cch.org.tw (S.-H.L.); 99257@cch.org.tw (B.-C.J.); 13081@cch.org.tw (M.-T.L.); 80690@cch.org.tw (W.-H.C.); 2Institute of Genomics and Bioinformatics, National Chung Hsing University, Taichung 402, Taiwan; 3Ph.D. Program in Translational Medicine, National Chung Hsing University, Taichung 402, Taiwan; 4Department of Recreation and Holistic Wellness, MingDao University, Changhua 523, Taiwan; 5Thoracic Medicine Research Center, Changhua Christian Hospital, Changhua 500, Taiwan; 181065@cch.org.tw; 6Big Data Center, Changhua Christian Hospital, Changhua 500, Taiwan; 179297@cch.org.tw; 7Graduate Institute of Statistics and Information Science, National Changhua University of Education, Changhua 500, Taiwan

**Keywords:** asthma, fungal sensitization, IL-17A, *Candida albicans*, mediation effect

## Abstract

Sensitization to fungal allergens is one of the proposed phenotypes in asthma. An association between fungal sensitization and worse clinical outcomes is apparent. Moreover, fungal sensitization in asthma that is associated with different type of immunological mechanism has been reported. How the role of cytokines mediates the association between fungal sensitization and poorer asthmatic outcomes remains unclear. We aimed to determine role of cytokines in the relationship between fungal sensitization and worse clinical outcomes in asthma. Method: We conducted a prospective study to recruit adult patients with asthma. Data including age, sex, height, weight, smoking history, medication, emergency visit and admission, pulmonary function testing result, and Asthma Control Test (ACT) scores were collected. We used the automated BioIC method to measure fungal allergen sIgE in sera. Serum levels of Interleukin (IL) -4, IL-13, IL-6, IL-9, IL-10, IL-17 A, IL-22, Interferon (IFN) -γ, Immunoglobulin E (IgE), Tumor necrosis factor-α (TNF-α), and Transforming growth factor-β (TGF-β) were measured using ELISA. Result: IL-6 and IL-17A had a significant positive correlation between sensitization and most fungi species compared to IgE. Sensitization to *Candida albicans* had strongly positive association both with IL-6 and IL-17A. However, only IL-17A had significant relationship with ED visit times. The mediation analysis result indicates that IL-17A had a significant positively mediating effect (ME) on the association between *Candida albicans* and ED visit times. Conclusion: IL-17A is a potential mediator to link *Candida albicans* sensitization and ED visits for asthma. We suggest that patients with fungal sensitization, such as *Candida albicans*, have poorer outcomes associated with Th17-mediated immune response rather than Th2.

## 1. Introduction

Asthma is a chronic inflammatory disease of the airways that significantly impairs quality of life [1]. It is also characterized by a variety of clinical presentations and outcomes, which can be classified into different phenotypes [2]. A previous study has suggested that sensitization to fungal allergens is one of the proposed phenotypes [3]. Cumulative studies demonstrate that fungal sensitization in patients with asthma has been associated with increased asthma severity as well as worse clinical outcomes, including worse asthma control, decreased lung function, increased hospital and intensive care unit (ICU) admissions, respiratory arrest, and asthma-related deaths [4,5,6,7,8]. Although an association between fungal sensitization and worse clinical outcomes is apparent, whether such an association is causal remains unconfirmed.

Fungal sensitization is an immune-mediated response to a fungus without evidence of inflammation or tissue damage [9]. In previous studies, both innate and adaptive immunity related to fungal sensitization in asthma. A review study indicated that allergic sensitization to fungi is mediated by the innate immune response driven by the innate lymphoid cells group 2 and the adaptive immune response driven by TH2 cells [4]. However, immune responses to fungi are mediated by a network of innate and adaptive immune cells, including but not limited to ILC2s and Th2 cells. Fairs A et al. show elevated levels of neutrophils in *A. fumigatus*-IgE–sensitized patients in comparison to non-sensitized patients with asthma, suggesting a Th1- or Th17-mediated immune response [6]. Moreover, an animal model of fungal-sensitized asthma found that IL-1Ra deficiency enhanced Th1 and Th17 immunity, increased neutrophil recruitment, and exacerbated disease [10,11]. Furthermore, a murine acute allergic asthma model demonstrated that sensitization with *A. fumigatus cpe* also elicited a higher percentage of IL-17AF+ eosinophil cells compared with OVA sensitization [12]. These studies reflected that fungal sensitization in asthma are associated with different type of immunological mechanism, including neutrophil, eosinophil, Th1, Th2, and Th17.

Cytokines, or intracellular signaling proteins, target specific cells causing consequences such as cell-mediated immunity and allergic responses [13,14,15]. Therefore, cytokines were chosen in order to investigate known cytokines of interest in asthma and to also examine the complex groups of Th1, Th2, Tregs, and Th17 representative cytokines. A previous study suggested that Th1, Th2, and Th17 immune responses relate to fungal sensitization in asthma; however, the cytokine profile of asthmatic patients who are sensitized to fungi is rare. Recent study reported that interleukin-33 levels were higher in severe asthmatic patients with fungal sensitization than in those without fungal sensitization. However, the study only analyzed the association between multiple fungal sensitization and cytokine, the relationship between species-specific sensitization to fungi and cytokines was not revealed [16].

To understand the immunological mechanism between fungal sensitization and asthma is vital for devising therapeutic interventions to prevent worse outcomes. In this study, we aimed to determine role of cytokines in the relationship between fungal sensitization and worse clinical outcomes in asthma. Firstly, we analyzed the cytokine profile regarding Th1, Th2, and Th17 immune response associated with sensitization to specific fungi species. Secondly, we evaluate the role of cytokine in the association between fungal sensitization and measures of disease control and worsen outcomes in asthma.

## 2. Material and Methods

### 2.1. Study Population

We enrolled patients who were at least 20 years old and presented with asthma; they were recruited from the Changhua Christian Hospital (Changhua, Taiwan) from 2012 through 2019. Patients with pulmonary tuberculosis, bronchiectasis, chronic obstructive pulmonary disease, chronic bronchitis, lung cancer, and cystic fibrosis were excluded from the study. The living environment of the patients registered this study was similar to that of patients who lived in nearly the same area (urban area around Changhua in Taiwan). Therefore, there was no major difference in climate, environment, etc., among patients. The study protocol was approved by the Institutional Review Board of Changhua Christian Hospital (IRB number 120607; approval date: 20/11/2012). The protocol was implemented in compliance with the Declaration of Helsinki. All participants provided written informed consent.

### 2.2. Collection of Clinical Information

Data including age, sex, height, weight, and smoking history were obtained from a chart review. Medication use and health care utilization, including emergency visits and admissions in the year, were obtained from medical records. The control status of asthma and disease-specific quality of life were determined using Chinese version of the Asthma Control Test (ACT).

### 2.3. Pulmonary Function Assessment

Pulmonary function, during stable asthma, was measured using a spirometer, which accommodates criteria of the American Thoracic Society. Each participant was required to perform three cycles of inhalation and exhalation, and the best forced expiratory volume in one second (FEV1), PEF, and forced vital capacity (FVC) were selected. GLI-2012-predicted values were used as the reference values for the executed spirometry.

### 2.4. Detection of Allergen Sensitization

We used the automated microfluidic-based multiplexed immunoassay system (BioIC Allergen-Specific IgE Detection Kit; GENERAL MEDICAL CO., LTD, Taichung, Taiwan) to measure fungal allergen sIgE in sera. The levels equal to or greater than Class 1 (≥1 AU) were considered positive. The sIgE to fungi allergens, including *Aspergillus flavus, Aspergillus niger, Penicillium chrysogenum, Botrytis cinerea, Fusarium solani, Rhodotoeula, Trichophyton rubrum, Saccharomyces, Penicillium, Penicillium notatum, Cladosporium herbarum, Aspergillus fumigatus, Candida albicans,* and *Alternaria alternata* were detected.

### 2.5. Serum Levels of Cytokines

Serum levels of IL-4, IL-13, IL-6, IL-9, IL-10, IL-17 A, IL-22, IFN-γ, TNF-α, and TGF-β measured using The BioLegend LEGEND MAX™ Enzyme-Linked Immunosorbent Assay kits (Biolegend, San Diego, CA, USA). IL-19 was detected using Quantikine Enzyme-Linked Immunosorbent Assay kits (R&D Systems, Minneapolis, MN, USA).

### 2.6. Statistical Analysis

Statistical analysis was performed using SPSS 22.0 for Windows (IBM Corporation, Armonk, New York, NY, USA). Data are expressed as mean ± SD, median, or percentage. Two-way comparisons were performed using the unpaired t-test for parametric variables and the Mann–Whitney test for nonparametric variables. Categorical variables were compared using the χ^2^ test or Fisher’s exact test. Spearman correlation analysis were used to determine the relationships between fungi, clinical outcomes, and cytokines. Correlation matrix plot were produced in R using the ggplot2 package. We performed mediation analysis using the SPSS PROCESS macro, version 2.16 (model 4), developed by Hayes [17]. All statistical tests were two-sided, with *p* < 0.05 considered to indicate statistical significance.

## 3. Results

### 3.1. Clinical Features of Patients with Asthma

As presented in Table 1, 97 patients (42.86% men) with asthma were enrolled in the study. The mean age was 56.71 ± 12.62 years old. In total, 84.5% of patients were never smokers, and 57.7% of patients had steroid use. The mean of ACT score, FEV1% predicted, and FEV1/FVC were 20.3 ± 3.9, 71.3 ± 19.3, and 72.2 ± 12.9, respectively. Additionally, 7.2% of patients experienced an ED visit and 12.4% of patients had admission.

### 3.2. Correlation among Fungi, Clinical Outcome of Asthmatic Patients, and Cytokines

The association between fungal sensitization and worse asthmatic outcomes has previously been reported. We first examined the relationship between fungi species and clinical outcomes. The correlation analysis results found that no significant relationship between fungi species and asthmatic outcomes, including pulmonary function testing result, ACT score, steroid use, ED visit times, and admission times (Figure 1A). Next, we evaluated the association between fungi species and immune markers. The results demonstrated that IL-6 and IL-17A have a positive relation with most fungi species. IL-6 was especially positively associated with *Aspergillus flavus, Aspergillus niger, Botrytis cinerea, Trichophyton rubrum, Cladosporium herbarum, Aspergillus fumigatus*, and *Candida albicans*. IL-17 had a particularly positive association with *Botrytis cinerea, Saccharomyces*, and *Candida albicans* (Figure 1B). However, only IL-17 was positively associated with ED visit times and FEV1 (Figure 1C).

### 3.3. Correlation among Botrytis cinerea, Saccharomyces, and Candida albicans and ED Times and IL-17A

A strong positive correlation was observed between IL-17A and *Botrytis cinerea*, *Candida albicans*, and *Saccharomyces* (*Botrytis cinerea*: r = 0.34, *p* < 0.0001; *Candida albicans*: r = 0.36, *p* < 0.0001; *Saccharomyces*: r = 0.39, *p* < 0.0001). The correlation between IL-17A and ED visit times was positively significant. ED visit times had no significant correlation with *Botrytis cinerea, Candida albicans, and Saccharomyces* (Figure 2).

### 3.4. IL-17A Levels in Asthmatic Patients with and without Fungal Sensitization Grouped by ED Visit Times

Previous studies indicated that fungal sensitization is associated with increased asthma severity and poorer clinical outcomes. However, we only observed a correlation between IL-17A and sensitized to various fungi (*Botrytis cinerea, Saccharomyces*, and *Candida albicans*), as well as between ED visit times and IL-17A, but not between fungal sensitization and ED visit times. Thus, we further investigated the IL-17A Levels in asthmatic patients with or without fungal sensitization stratified by ED visit times. The results show that patients who were sensitized to *Botrytis cinerea* had higher levels of IL-17 than patients without sensitization to *Botrytis cinerea* (Figure 3A). Patients with *Saccharomyces* had no ED visit event (Figure 3B). Conversely, patients with an ED visit (1 time) sensitized to *Candida albicans* had higher IL-17A compared to patients who were not sensitized, but without significant difference (Figure 3C).

### 3.5. Role of IL-17A in the Associations between Candida albicans and ED Visit Times

To clarify the role of IL-17A in the relationship between *Candida albicans* sensitization and ED visit times. We investigated the role played by IL-17A in transmitting *Candida albicans* sensitization changes to ED visit times in asthmatic patients using mediation analysis. The total effect of *Candida albicans* on ED visit times was −0.106 (95% CI: −0.454, 0.133). IL-17A had a significant positively mediating effect (ME) on the association between *Candida albicans* and ED visit times (ME = 0.0175, 95% CI: 0.012, 0.528) (Figure 4).

## 4. Discussion

Being sensitized to fungi is a potential risk factor for worsening asthma outcomes, but the mechanisms underlying the fungal sensitization associated with poorer asthmatic outcomes remain unclear. The present study shows a link between cytokines and fungal sensitization associated with poorer asthmatic outcomes. Firstly, we observed that IL-6 and IL-17A, but no IgE, had a significant association with sensitization to most species of fungi in out testing panel. Only ED visit times were positively associated with IL-17A, and we noticed that the level of IL-17A was higher in patients with *Candida albicans* sensitization who had ED visits compared with those who did not, though the difference is not significant. The mediation analysis revealed that IL-17A had positive mediation effect between *Candida albicans* sensitization and ED visit times. Overall, our finding indicated that sensitization to *Candida albicans* had a positive correlation with IL-17A levels, which were then associated with ED visits in asthmatics. Based on present results, IL-17A could be a biomarker for asthmatics with frequent emergency department visits for *Candida albicans* sensitization.

Sensitization to fungi is associated with increased asthma severity, poorer clinical outcomes, and mortality. In previous studies, the prevalence of fungal sensitization in asthma has varied. According to a UK study, 66% of patients with severe asthma were sensitized to one or more fungi, as determined by SPT or specific serum IgE testing or both [18]. Moreover, asthma patients referred to subspecialty clinics showed that 17.3% were allergic to fungi [19], but another study showed 76.3% to be allergic to at least one fungus [20]. Further, 32% of the 576 patients enrolled in the study with severe eosinophilic asthma demonstrated sensitivity to fungal allergens [21]. In our study, we found that 92.8% of asthmatic patients had fungal sensitization. These results indicate that the exact prevalence of fungal sensitization in patients with asthma remain unclear. It is likely that the large differences seen in the prevalence of fungal sensitization among asthmatic patients are related to differences in patient populations, testing methods, and geographic locations.

Fungal sensitization is associated with worse asthmatic outcomes being reported. For example, the number asthmatic patients with fungus sensitization requiring intensive care unit admission and mechanical ventilation was higher than those without fungus sensitization or nonfungal sensitization [19]. Furthermore, the lung function of patients sensitized to thermotolerant filamentous fungi was lower than that of patients not sensitized to any fungi [20]. Despite this, we were unable to find any association between fungal sensitization with measures of control, severity, or steroid use in our population. However, we found that patients with fungal sensitization had a higher prevalence of ED visits than patients without fungal sensitization (7.78% versus 0%, data not shown). These finding suggest that exposure to inhaled allergens to which patients are sensitized can increase asthma symptoms or precipitate exacerbation. According to GINA guidelines, patients with persistent symptoms and/or exacerbations should undergo allergen testing. It is possible that patients with fungal sensitization were not identified because of lack of testing [19]. Thus, greater awareness of asthmatics at risk for more difficult disease outcomes should support earlier identification of fungal sensitization.

Allergic asthma is defined by the presence of allergic sensitization and a correlation between allergen exposure and asthma symptoms. One biomarker of allergic asthma is total serum IgE level, which is more commonly elevated in allergic compared with nonallergic asthma [22], is inversely associated with lung function in asthmatics [23], and is associated with the prevalence of asthma [24]. A previous study reported that patients with fungal sensitization (*Penicillium chrysogenum*, *Cladosporium herbarum*, *Aspergillus fumigatus*, *Mucor racemosus*, *Stemphylium herbarum*, and *Alternaria alternata*) had a higher total serum IgE concentration than patients with no sensitization or nonfungal sensitization [19]. In present study, our data demonstrate that sensitization to most species of fungi had a positive relationship with IL-6 and IL-17 rather than IgE. In previous study, eosinophils and eosinophilic production of IL-23 and IL-17 were shown to be beneficial in invasive aspergillosis but detrimental in allergic disease in a mice model [12]. In addition, IL-6 is essential for mucus hypersecretion by airway epithelial cells triggered in response to inhaled *Aspergillus fumigatus* extract, which was found in a mouse model of allergic airway inflammation induced by direct airway exposure to extracts of *Aspergillus fumigatus* [25]. These findings suggest that the immune response regarding fungal sensitization is not only associated with IgE, but also other mechanisms involved that are related to IL-6/IL-17A axis. In allergic asthma, B cells may regulate the T cell response by modulating the phenotypic response. For example, in a JH^−/−^ (B cell-deficient) murine model of fungal allergic asthma, levels of the inflammatory cytokines IL-6 and IL-17A were significantly elevated, and there was significantly more robust airway eosinophilia and neutrophilia [26]. The evidence suggests that the IL-6/IL-17A axis is associated with fungal allergic asthma in conditions with B cell deficiency. Thus, the IL-6/IL-17A axis would be a possible mechanism associated with fungal sensitization other than IgE.

Accumulating evidence now suggests that Th17 cells and their related cytokines are also involved in the pathophysiology of allergic asthma. IL-17 expression is increased in the lung, sputum, bronchoalveolar lavage fluid (BALF), and sera in patients with asthma, and the severity of AHR is positively correlated with IL-17 expression levels [27,28]. In our study, IL-17, but not IL-6, had a positive correlation with ED visits and had a significant mediation effect on the association between *Candida albicans* sensitization and ED visit. A previous study found that IL-17 increased in *Aspergillus fumigatus*-sensitized mice [29]. In addition, IL-17A response is associated strongly with acute ABPA, and its specific decline in response to ABPA treatment suggests that this atypical Th17 response plays an active role in manifesting and/or exacerbating disease [30]. The evidence supports the hypothesis that IL-17A has a mediated effect on the relationship between *Candida albicans* sensitization and poorer clinical outcomes, particularly ED visits. A previous study identified the role of IL-17-mediated immunity in Candidiasis, and the implications for clinical therapies for both autoimmune conditions and fungal infections [31]. In asthma, to our best knowledge, our findings firstly provide evidence that IL-17A is a potential mediator to link *Candida albicans* sensitization and poorer clinical outcomes. Despite the underlying cell-mediated mechanism is unclear, a study report that protective lung Th2 and Th17 cell responses against the common mucosa-associated fungus *Candida albicans* are coordinated through lung megakaryocytes and platelets [32]. The findings pointed suggested that Th17, an IL-17-producing cell, may have a protective effect against allergy to *Candida albicans* in the lungs.

Th17 immune responses differ between the sexes due to suppressive and enhancing effects of sex hormones [33]. A previous study indicated testosterone is associated with expansion of Th17 populations in a murine model of experimental autoimmune encephalomyelitis (EAE) [34]. However, a recent study indicated that the increased production of IL-17A by 17β-estradiol and progesterone from TH17 cells may provide a potential mechanism for the increased prevalence of severe asthma in women compared with men [35]. In contrast, estrus levels of estradiol downregulated the Th17 response to *Candida albicans* in in vivo vaginal infection models [36]. In our finding, we found that the IL-17 levels in asthmatic patients were 3.28 and 4.17 in males and females, respectively, but there is no significance between sex differences (*p* = 0.369) (Appendix A). The correlation between IL-17 and *Candida albicans* was 0.233 and 0.458 in males and females, respectively, but there are no significant differences between males and females in the correlation between IL-17 and *Candida albicans* (*p*_interaction_ = 0.166) (Appendix A). The finding demonstrates that sex difference has no influence on the correlation between IL-17 and *Candida albicans* in our population. Taken together, we suggest that the influence of sex difference on IL-17 is inconsistent that may depend on disease-specific characteristics or severity; thus, the mediation effect of IL-17 between poor outcomes and *Candida albicans* across sex difference is needs further study.

There are some limitations of our study. Firstly, further evaluation in a multiple center setting is needed to expand the generalizability. Secondly, we lacked information regarding immune cell type; therefore, the cell-mediated immune response related to IL-17A in fungal sensitization, especially in *Candida albicans*, needs to be further investigation. Thirdly, since some of our asthma patients were without ED visits, the causal effect is difficult to clarify due to limitation of statistical method.

## 5. Conclusions

In present study, we found that both IL-6 and IL-17A have a strong positive correlation with *Candida albicans*. Only IL-17A had significant positive association with ED visit times. There is a significant mediation effect of IL-17A on the association between *Candida albicans* sensitization and ED visit times. Altogether, IL-17A is a potential mediator to link *Candida albicans* sensitization and ED visits for asthma. Patients with fungal sensitization, such as *Candida albicans*, who have worse outcomes may associate with Th17-mediated immune responses rather than Th2.

## Figures and Tables

**Figure 1 biomedicines-10-01452-f001:**
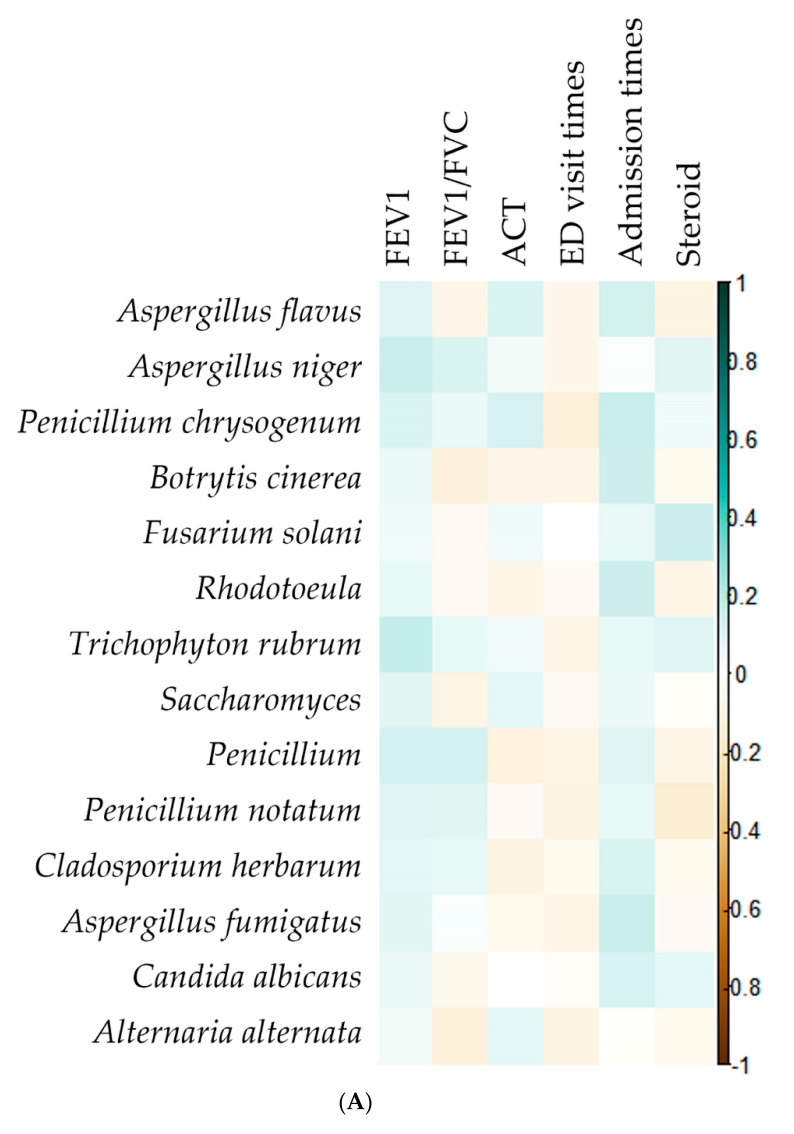
Correlation matrix plot between (**A**) fungus and asthma-related outcomes, (**B**) inflammatory cytokine and asthma-related outcomes, and (**C**) fungus and inflammatory cytokine. * *p* < 0.05, ** *p* < 0.01, *** *p* < 0.001.

**Figure 2 biomedicines-10-01452-f002:**
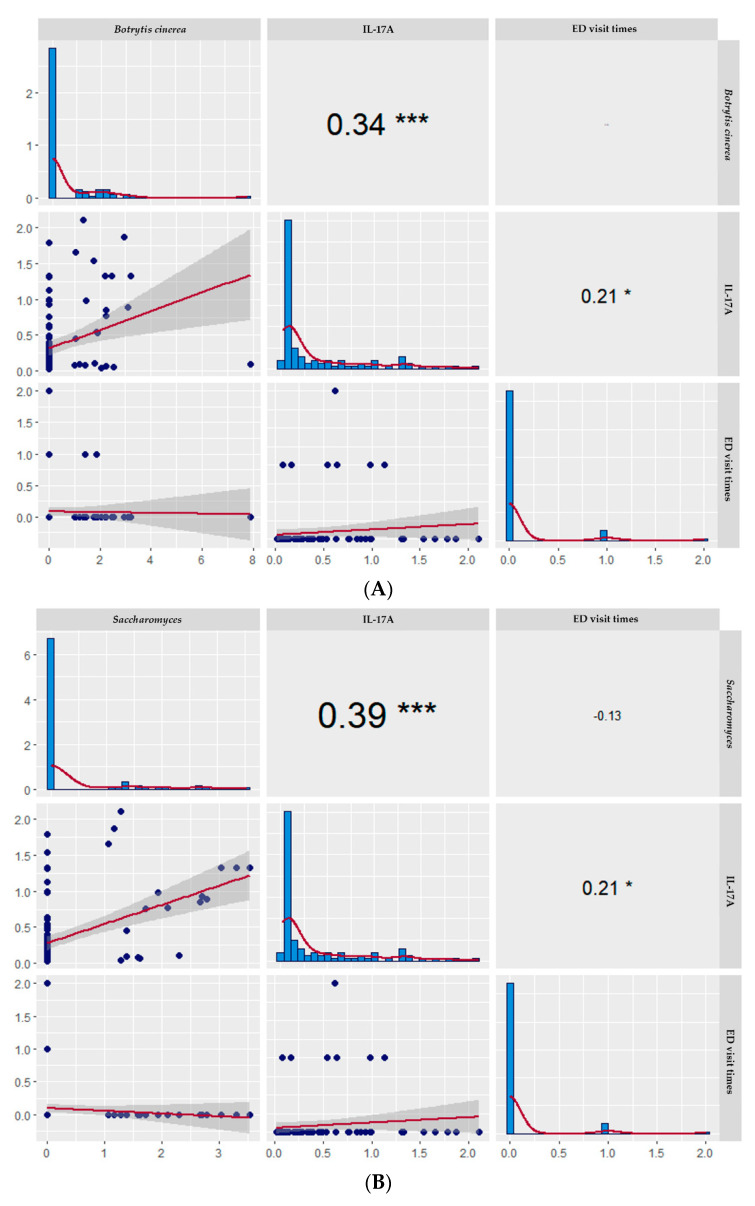
The scatter matrix, histogram, and Spearman rank correlation matrix among (**A**) *Botrytis cinerea*, (**B**) *Saccharomyces*, and (**C**) *Candida albicans* and ED times and IL-17A. * *p* < 0.05, *** *p* < 0.001.

**Figure 3 biomedicines-10-01452-f003:**
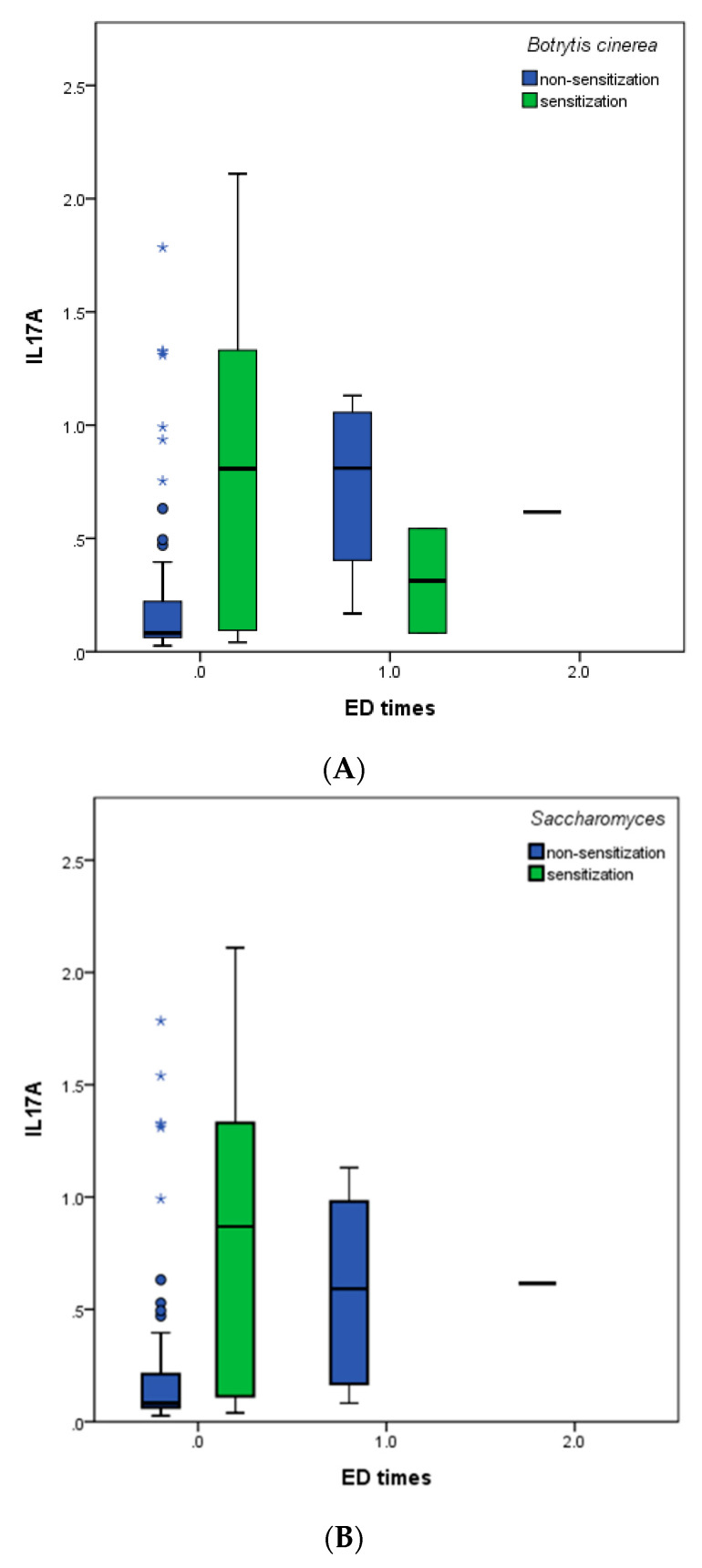
Comparing IL-17A levels in asthmatic patients with fungal sensitization and non-sensitization grouped by ED visit times. *Botrytis cinerea* (**A**); *Saccharomyces* (**B**); *Candida albicans* (**C**). * *p* < 0.05.

**Figure 4 biomedicines-10-01452-f004:**
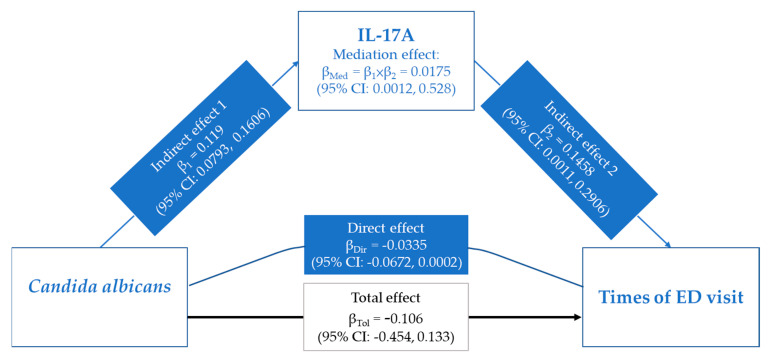
The contribution of IL-17A for the association between *Candida albicans* and ED visits. Single mediation models and regression coefficients (β), with 95% confidence intervals (95% CI) examining potential mediators of IL-17A.

**Table 1 biomedicines-10-01452-t001:** Clinical features of patients with asthma.

	Patients With Asthma (N = 97)
Age (mean ± SD)	49.1 ± 18.2
BMI (kg/m2) (mean ± SD)	24.8 ± 4.2
Gender (N, %)	
Male	39 (40.2%)
Female	58 (59.8%)
Smoking status	
Never smoker	82 (84.5%)
Ever smoker	5 (5.2%)
Current smoker	10 (10.3%)
Steroid use	
No	41 (42.3%)
Yes	56 (57.7%)
ACT (mean ± SD)	20.3 ± 3.9
FEV1% (mean ± SD)	71.3 ± 19.3
FEV1/FVC (mean ± SD)	72.2 ± 12.9
ED visit times (N, %)	
0 times	90 (92.8%)
1 times	6 (6.2%)
2 times	1 (1.0%)
Admission times (N, %)	
0 times	85 (87.6%)
1 times	9 (9.3%)
2 times	3 (3.1%)
Fungal sensitization (N,%)	
Without sensitization	7 (7.2%)
With sensitization	90 (92.8%)
*Aspergillus flavus* (mean ± SD)	0.4 ± 0.82
*Aspergillus niger* (mean ± SD)	1.26 ± 1.72
*Penicillium chrysogenum* (mean ± SD)	0.66 ± 3.32
*Botrytis cinerea* (mean ± SD)	3.66 ± 31.35
*Fusarium solani* (mean ± SD)	1.34 ± 1.32
*Rhodotoeula* (mean ± SD)	2.03 ± 3.46
*Trichophyton rubrum* (mean ± SD)	1.12 ± 1.95
*Saccharomyces* (mean ± SD)	0.38 ± 0.86
*Penicillium* (mean ± SD)	0.22 ± 1
*Penicillium notatum* (mean ± SD)	0.2 ± 0.6
*Cladosporium herbarum* (mean ± SD)	1.09 ± 3.41
*Aspergillus fumigatus* (mean ± SD)	1.6 ± 3.45
*Candida albicans* (mean ± SD)	2.23 ± 2.15
*Alternaria alternata* (mean ± SD)	0.19 ± 0.6
IgE (mean ± SD)	394.08 ± 947.1
IL-4 (0.0125–1 ng/mL) (mean ± SD)	0.025 ± 0.096
IL-6 (ng/mL) (mean ± SD)	0.446 ± 0.503
IL-9 (0.04–3 ng/mL) (mean ± SD)	0.258 ± 0.386
IL-10 (0.04–2.5 ng/mL) (mean ± SD)	0.05 ± 0.089
IL-17 A (0.03125–2 ng/mL) (mean ± SD)	0.381 ± 0.501
IL-13 (0.0625–4 ng/mL) (mean ± SD)	0.137 ± 0.394
IL-19 (0.0625–2 ng/mL) (mean ± SD)	0.195 ± 0.147
IL-22 (0.0125–1 ng/mL) (mean ± SD)	0.023 ± 0.092
IFN-γ (0.047–1.5 ng/mL) (mean ± SD)	0.209 ± 0.462
TGF-β (0.03–2 ng/mL) (mean ± SD)	0.091 ± 0.079
TNF-α (0.03125–2 ng/mL) (mean ± SD)	0.408 ± 0.602

## Data Availability

Due to ethical reasons, these data cannot be made public.

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
