# Peer review of "The Mediating Effect of Cytokines on the Association between Fungal Sensitization and Poor Clinical Outcome in Asthma"

_biomedicines, 2022, doi:10.3390/biomedicines10061452_

Round 1

Reviewer 1 Report

Paragraph: 3.3. Correlation among Botrytis cinerea, Saccharomyces and Candida albicans, ED times, and IL- 17A  ....." 

... ED visit times are positively correlated with Botrytis cinerea (r = 0.023, p=0.820) and negatively correlated with Candida albicans and Saccharomyces, respectively.  (Candida albicans: r = -0.05, p=0.623; Saccharomyces: r = -0.13, p=0.197).

The claims in this paragraph are not clear and need to be corrected.

Author Response

Thank you for your valuable comments on our manuscript. We would like to express our sincere appreciation for your professional and insightful remarks on our manuscript. We have addressed all of the suggestions made by Reviewer 1. Please see the attachment.

Reviewer 2 Report

Lin et al. studied the association between specific sensitization for fungi and cytokines and clinically identified the relationship with poor outcomes such as emergency department visits in asthma. The authors raised the importance of cytokines associated with fungi sensitization in assessing subtypes of asthma and further provided therapeutic clues. Th17 is generally more likely to be expressed in men than in women due to hormonal effects, so it would be great to address where there is an effect modifier according to men and women in this study. In the 208th line, it would be correct to change ‘is vary’ to ‘varies.’ Please check some minor types and expressions for abbreviations throughout the manuscript. 

Author Response

Thank you for your valuable comments on our manuscript. We would like to express our sincere appreciation for your professional and insightful remarks on our manuscript. We have addressed all of the suggestions made by Reviewer 2. Please see the attachment.
